# *AdaSR* : Adaptive Super Resolution for Cross Platform and Dynamic Runtime Environments

## Abstract

Image super resolution models (SR) have shown great capability in improving the visual quality for low-resolution images. Due to the compute and memory budgets of diverse platforms, e.g., cloud and edge devices, practitioners and researchers have to either (1) design different architectures and/or (2) compress the same model to different levels. However, even on the same hardware, the compute resource dynamics change due to other running applications, meaning that one single model that satisfies required frames-per-second (FPS) when executed in isolation may not be suitable when other running applications present. To overcome the problem of requiring custom model design and real-time resource availability changes, we propose *AdaSR* , an `Adaptive` SR framework via shared architecture and weights for cross platform deployment and dynamic runtime environment. Unlike other works in literature, our work focuses on the development of multiple models within a larger meta-graph such that they can fulfill latency requirements by compromising as little performance as possible. Particularly, *AdaSR* can be used to (1) customize architectures for different hardware and (2) adaptively change the compute graph in dynamic runtime environment with no extra cost on memory and/or storage. We extensively test *AdaSR* on different block-based GAN models, and demonstrate that *AdaSR* can maintain Pareto optimal performance for latency vs. performance tradeoff in comparison with state-of-the-art with much smaller memory footprint and support dynamic runtime environments.

## 1 Introduction

Image super-resolution models (SR) improve visual quality of low-resolution images and have been widely used in many applications such as video streaming, compression, image recovery, enhancement Park et al. (2003). Driven by recent advances in deep neural network models (DNN) Dong et al. (2015; 2016); Bashir et al. (2021), the performance of SR has been greatly improved. To use SR models in real world environment, such as cloud and edge devices, it needs to meet Quality of Service (QoS) standards Zhang et al. (2020), e.g., maintaining a minimum frames per second (FPS) to provide a smooth perceived visual experience. Given the high computation and memory demands of SR models Chen et al. (2022a), practitioners and researchers usually need to customize architectures for different platforms to meet QoS while achieving the high performance. In addition, the available computing and memory resources may dynamically change due to other running applications, which requires SR models to be adaptive in real-time.

To provide cross platform deployment support, earlier works handcraft efficient DNN architectures for different hardware Lee et al. (2019); Ma et al. (2019); Dong et al. (2018). Later works employ compression schemes such as quantization Ayazoglu (2021); Hong et al. (2022), pruning Jiang et al. (2021); Zhang et al. (2021c), and Knowledge Distillation (KD) Gao et al. (2018); Aguinaldo et al. (2019) to reduce model sizes. However, these methods only produce singular models for certain devices, or require significant hand-tuning efforts to generate a set of models for different devices. Recent works explore Neural Architecture Search (NAS) Fu et al. (2020); Bashir et al. (2021) to automate the model generation process for different hardware. However, NAS usually requires costly time and resources to come up with good search space and search proper architectures for each hardware Ren et al. (2021). More importantly, none of the above works addresses the challenges in dynamic runtime environment where SR models need to swiftly adapt to dynamically changing resources without high memory cost.

To address the aforementioned issues, we propose *AdaSR* , an `Adaptive SR` framework via shared architecture and weights for cross platform deployment and dynamic runtime environment. Our key insight here is to adaptively change the depth and the channel size of SR models with shared weights and architecture so that SR models can adapt quickly with little extra memory cost. We achieve this by employing a *progressive knowledge distillation* approach to train the size/compute-adaptive models in a layer-wise manner. Such a function matched training enables improved consistency in learning representation between the original and adapted models. However, performing progressive knowledge distillation on SR models is non-trivial because the large variety of blocks makes it impractical to hand tune each. To stabilize the training of *AdaSR* such that it is robust to dynamic changes in operations, we propose a progressive approach to derive loss functions for each block and function matching operations with max-norm regularization to address dimension mismatches.

Thanks to the above design, *AdaSR* can distill knowledge with different compression levels for different hardware (e.g., different security cameras), and also support adaptively change the compute graph in dynamic runtime environments (e.g., mobile phones with concurrently running applications). Prior works tend to propose a single model for each target hardware Kim et al. (2016); Tai et al. (2017); Nie et al. (2021); Wu et al. (2021) while maximizing performance. Instead of aiming to beat the state-of-the-art, we focus on generating multiple models that are close to the Pareto frontier for the performance vs. latency tradeoff. This allows us to have competitive model performance while still being able to maintain a minimum FPS for a given set of resources. We perform extensive evaluation by comparing *AdaSR* with popular efficient SR models CARN Ahn et al. (2018), ESRGAN Wang et al. (2018b), RCAN Zhang et al. (2018a) and the state-of-the-art FMEN model from the 2022 NTIRE challenges NJU_JET team Li et al. (2022). These models cover a wide range of inference latencies and development techniques (i.e. hand-designed, compression and Neural Architecture Search). We deploy our models on real hardware systems such as mobile, laptop and GPUs for evaluation. The results show that *AdaSR* achieves Pareto frontier of the prior arts while having 80% smaller memory footprint and can adapt to dynamically available resources in dynamic runtime environments with little overhead.

## 2 RELATED WORKS

**Efficient SR models.** There is a rich set of works use novel architectural properties to hand tune SR models. For example, a line of work attempts reduction by using more efficient computation blocks such as DRCN and DRRN Kim et al. (2016); Tai et al. (2017) using cheaper recursive layers, GhostSR Nie et al. (2021) for lightweight residual blocks and SESR Cheng et al. (2018) replacing standard convolutions with collapsible linear blocks. Another line of works Mei et al. (2021); Liu et al. (2022) exploit attention mechanism for efficient representation capacities of SR models. Other papers use a variety of techniques such as splitting channels more efficiently Hui et al. (2018) and shifting the position of the upsampling operator Dong et al. (2016). The winner of the PIRM challenge Blau et al. (2018), MobiSR Lee et al. (2019), and other related works Li et al. (2022); Chan et al. (2022); Angarano et al. (2022) also apply similar methods. The state-of-the-art performance in SR is Gao & Zhou (2023), which proposes a combination of frequency grouping fusion blocks, multi-way attention blocks, lightweight residual concatenation blocks and novel convolutional structures. However, these SR models are usually hand-designed models customized to a specific hardware. Moreover, there is a strong assumption that these models will have full use of all resources at all times, which is not the case for real world systems such as mobile devices where multiple applications may share resources.

**SR Compression.** Another line of works aim to reduce the operations and parameters of existing models using compression techniques such as pruning Liu et al. (2020), quantization Gholami et al. (2021), and knowledge distillation Wang & Yoon (2021). Recent SR pruning works include SMSR Wang et al. (2021), ASSLN Zhang et al. (2021b), and SRPN Zhang et al. (2021c), which employ techniques such as learning sparse masks to prune redundant operations during inference, while works such as Ayazoglu (2021), Hong et al. (2022) and Xin et al. (2020) use advanced quantization methods. Knowledge Distillation has also been used to compress SR models such as the work in Gao et al. (2018); Zhang et al. (2021a); Suresh et al. (2022); Chen et al. (2022b) However, all these works focus on generating single models. As such, they are not suitable for cross platform deployment or dynamic resource adaptability. We compare our *AdaSR* with Lee et al. (2019); Du et al. (2022); Dong et al. (2014); Wang et al. (2018b); Ahn et al. (2018); Zhang et al. (2018b;a) in our evaluation as they reside at various points of the state-of-the-art in latency vs. performance tradeoff and can be used to bound the Pareto frontier. Neural Architecture Search (NAS) Ren et al. (2021) has been

explored to find efficient, lightweight, and accurate SR models. Works such as Tri-level NAS Wu et al. (2021), AutoGAN Distiller Fu et al. (2020) and DARTS Liu et al. (2018) have been used to derive compact model versions. Specific NAS-based works FALSR Chu et al. (2021), DeCoNAS Ahn & Cho (2021) and MoreMNAS Chu et al. (2020) have been applied for model compression for real-time SR on mobile devices. However, these methods are expensive, require customized searches and do not support dynamic run-time latencies. We compare our *AdaSR* with the Tri-level, AutoGAN, and FALSR NAS approaches in our evaluation.

## 3 PROPOSED METHOD

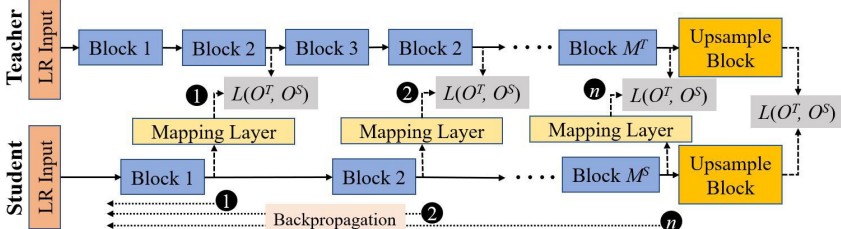

Figure 1: *AdaSR* **Model Architecture.** This figure shows the steps for progressive knowledge distilled training. Every $m^{th}$ block of the original model is distilled to the adaptable model's block. The backpropagation starts from the current distilled block to the block at the beginning. $L$ is the loss function using output values of $O^T$ and $O^S$ from the original and adapted models, respectively.

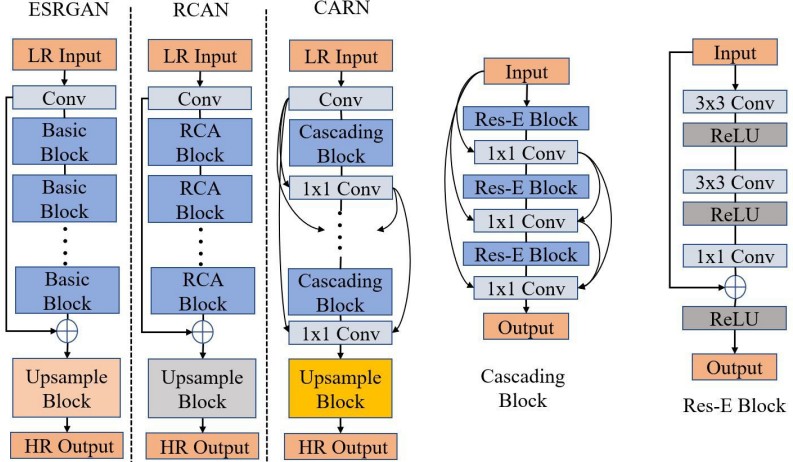

Figure 2: **Network diagrams of CARN, RCAN, and ESRGAN models.** They use repetitive blocks stacked on top of each other with residual connections. Cascading and Res-E block architectures from CARN are shown in detail here as an example of how the blocks of these models use the convolutional operations.

To support cross platform deployment and dynamic runtime environment of SR models, our key insight is to share architecture and weights to enable quick model adaption while reducing memory consumption. Specifically, the model is adapted by changing the depth and the channel size to meet the QoS requirements under different resource constraints. However, direct reduction of depth and width usually results in significant performance degradation Hou & Kung (2022); Wang et al. (2018a). To address this challenge, we employ progressive knowledge distillation and loss function optimization to improve the performance of the adapted SR models. We use a loss function at each layer to minimize the output distribution discrepancy between the adapted model and the original model. This allows the adapted model to be consistent in learning representation of the original model. To derive the loss functions in an automated way, we introduce a Bayesian tuning method. We further introduce output matching operations with max-norm regularization to address the dimension mismatch issue between the original and adapted models.

### 3.1 OPERATION REDUCTION

First, we demonstrate how to precisely adapt the model size by changing the channel size and the number of blocks. Here we use the popular CARN model Ahn et al. (2018) as an example to

illustrate our idea, but it is generalizable to all block-based convolutional GANs. We denote the original model as $B^T$ and input size for any block as $H_{in}^i \times W_{in}^i \times C_{in}^i$, where $H$, $W$ and $C$ are the height, width, and channel dimensions respectively for block $i$. Here we leave out the bias and activation terms for simplicity. For a convolutional layer with kernel size $K$ and channel size $C_{out}$, the computational cost is:

$$K \cdot K \cdot C_{in}^i \cdot H_{out} \cdot W_{out} \cdot C_{out}, \tag{1}$$

where $H_{out}$ and $W_{out}$ are the output height and width dimensions respectively. For each block, the output dimensions are padded to keep them equal to the the input dimensions $H_{in} \times H_{out}$ since every block has the same set of operations. For simplicity and without loss of generalizability, we have $H_{in}^i = W_{in}^i = H_{out} = W_{out} = F$. As shown in Fig. 2, multiple $N$ convolutional operations are stacked to make a single block, and are sometimes followed by more operations such as identity convolutions, activation functions, and concatenation operations. We can compute their cost as a function $f_{cost}(F, C_{out})$, where $f$ is generalizable to a wide variety of operations in the block architecture. Thus, we can simplify the cost of a full block as:

$$\sum_1^N (K^2 \cdot F^2 \cdot C_{in} \cdot C_{out}) + f(F, C_{out}). \tag{2}$$

Because there are multiple blocks $M$ in an architecture followed by upsampling layers, which is denoted as $f_{up}$, we can derive a generalizable total cost function for the original model:

$$B_{cost}^T = \sum_1^M (\sum_1^N (K^2 \cdot F^2 \cdot C_{in} \cdot C_{out}) + f(F, C_{out})) + f_{up}(F, C_{out}). \tag{3}$$

If we consider the operation costs for $f, f_{up}$ are directly proportional to the input dimensions $F \times F \times C_{out}$, we can simplify the cost function as:

$$B_{cost}^T = C_{out} \cdot M(N \cdot K^2 \cdot F^2 \cdot C_{in} + f(F)) + f_{up}(F). \tag{4}$$

For the adapted model $B^S$, if we keep the overall architecture the same and only reduce the filter sizes $C_{out}$ across all the blocks with a ratio of $c$ and reduce the number of blocks $M$ to a ratio of $m$, we have the cost function of adapted model as follows:

$$B_{cost}^S = \frac{C_{out}}{c} \cdot \frac{M}{m}(N \cdot K^2 \cdot F^2 \cdot C_{in} + f(F)) + f_{up}(F), \tag{5}$$

which is $c \cdot m$ times less than $B^T$. A reduction in channel size (width) and number of blocks (depth) can have a proportional reduction in model size with little architecture re-engineering efforts.

## 3.2 *AdaSR* ARCHITECTURE DESIGN

Next, we explain how to utilize operation reduction to adapt SR models with the most profitable performance. Operation reduction may degrade the model performance Aguinaldo et al. (2019); Zhang et al. (2020); Gou et al. (2021) because training operation reduced models from scratch may not capture all the feature representations of the larger SR counterpart. To address this challenge, we propose a Knowledge Distillation (KD) Jin et al. (2019) based adaption scheme to optimize the performance of adapted models. The intuition behind using KD is that by using the already learned feature representations of larger SR models to teach the smaller adapted ones, we can preserve the performance while speedup the model adaption process. However, existing KD works for SR Gao et al. (2018); He et al. (2020); Angarano et al. (2022) mainly focus on matching the output distributions after a large number of layers, which falls short in maintaining consistent mapping of learned representations between the high and low dimensional spaces in the inner layers, even for networks with the same operation structures.

To better reflect the original models' feature representation in each layer of the adapted models in low dimensional space, we propose a new approach that can match output distributions at regular intervals. Specifically, we use the original SR model to *progressively* train the adaptable models layer-by-layer instead of simply training on the loss of the last layer's output distributions. The overall training method is shown in Fig. 1. As pointed out in Mirzadeh et al. (2020); Beyer et al. (2022); Jin et al. (2019), having such anchor points during training or *function matched training* can yield better performance for distilled models. This is because such a training procedure allows greater consistency between the original and distilled model's learning representations for the inner

layers. We introduce Bayesian Tuning Victoria & Maragatham (2021) to automatically get the appropriate loss function for each layer. We calculate the gradients based on this loss for each possible adapted models (i.e. every combination of width and depth possible) and then apply them together to make the trained weights robust to direct reductions in convolutional channels and blocks. To ensure the per-layer output dimensions of the smaller adaptable models to match with the original models, we further introduce function matching operations with *max-norm* regularization.

### 3.2.1 PROGRESSIVE KNOWLEDGE DISTILLATION

To simplify the illustration, we also use the CARN model Timofte et al. (2018); Agustsson & Timofte (2017) as an example, although the approach can be applied to any block-based convolutional GANs. We start building the adaptable student model by first choosing the lowest possible channel size and number of blocks (e.g., in CARN the smallest channel size is 8 and the smallest number of blocks is 1). We do this by ignoring the computation from the excluded blocks and throwing away the undesired channel dimensions when using the first block's output. This is done on the student's model, and the teacher's model is kept intact. We then take a batch of low-resolution input images, preprocess them and perform a forward pass to get the output distribution of the chosen block (i.e. block 1) from both original pre-trained teacher model and the untrained adaptable student model. These output distributions are then used to derive the gradients for the current student block via function matching (as explained in detail in the later sections). Once the gradients for this particular block is derived, we increase the size of the adaptable model in one dimension at a time (as shown step-by-step in Fig. 1), and repeat the process until we have derived the gradients for all numbers of blocks and channels adapted models at every available granularity level (we increase block size by 1 and channel size by 8 for every step). In this way, we *progressively* increase the adaptable model's size during the forward pass and derive gradients for each sub-model size.

### 3.2.2 FUNCTION MATCHING AND REGULARIZATION

At each block level, we get the output distributions of both the student and the teacher models. The idea here is to train the student such that the output of the block matches as much as possible with the teacher's corresponding block. This is done by calculating the student's gradients based on the loss function between the student's and teacher's output distributions. However, the reduced number of channels of adaptable models may cause the output dimensions of each block mismatch with those of the original models. To make the loss function work, the output distributions need to have the same dimensions/ We do this by adding another layer on top of the last layer of the adaptable model's block such that this mapping layer's output matches the original model's. This mapping layer is only used during the training of that layer and is dropped for layers already progressed from or during inference. The mapping layer can either be a linear layer which changes the output dimensions via matrix multiplication to match dimensions, or a convolutional layer with kernel and channel sizes equal to that of the original model's. We find convolutional layers work better since its learned feature representations are closer to the interpretations of the original model's convolutional layers.

While this method solves the matching problem between the original model and the adaptable model when calculating loss, it also results in a tandem training of the mapping layer weights and the adaptable model layer weights. The removal of the mapping layer during inference may yield worse performance. Here we employ *max-norm* regularization Srebro & Shraibman (2005) on the mapping layer to enforce upper bound of weights. The *max-norm* constraint $||w||_2 < c$ regulates the impact of that layer's weights and helps train the previous layers to closely represent the original model's output distributions. We find other regularization techniques such as *L2 norm* and *dropouts* are less effective because *max-norm* has a more direct bound on weights. We find $c = 0.0002$ is sufficient and we use this value for all experiments.

### 3.2.3 DEPTH CONSOLIDATION

One way to reduce parameters is to remove a block entirely. Since consistency and patient training helps performance Beyer et al. (2022), we perform knowledge distillation for each block on the adaptable model using the original model blocks. Ideally, the adaptable model has the same depth as the original model so that there is a one-to-one distillation mapping between each block. On removal of a block from the adaptable model, we re-adjust the anchor as $m = \lfloor \frac{M^T}{M^S} \rfloor$, where an adapted model with $M^S$ blocks gets distilled every $m$th layer from the original model's $M^T$, with the last block including the last leftover blocks. One issue here is that changing the number of blocks may cause dimension mismatch. For example, concatenation of the outputs of current and previous blocks is a common structural operation Ahn et al. (2018); Wang et al. (2018b), which results in that

dimensions depend on the number of blocks. In such cases, we also perform dimension matching using the aforementioned mapping layers.

---

**Algorithm 1** *AdaSR* Training

---

1: **Inputs:** Pre-trained original model $T$, dataset $Dataset$, adapted model to train $S$, training epochs $Epochs$, adapted model's depth and width options $depth$, $width$, loss function $L$.
2: **for** each $epoch$ in $Epochs$ **do**
3:   **for** each $batch$ in $Dataset$ **do**
4:     $global\_grads = []$
5:     $t_{1..N} = T(batch)$
6:     **for** each $D$ in $depth$ **do**
7:       $grads = []$
8:       **for** each $W$ in $width$ **do**
9:         $s_D^W = S(batch, W, D)$
10:         Depth Consolidation for anchoring $t_i$ and function Matching for $s_D^W$ to $t_i$
11:         $\lambda = BayesianOptimization(L)$
12:         $l = L(\lambda, s_D^W, t_i)$
13:         $grads.append(l.gradients())$
14:       **end for**
15:       $global\_grads.append(grads)$
16:     **end for**
17:     $S.update(grads)$
18:   **end for**
19: **end for**

---

### 3.2.4 BAYESIAN-TUNED LOSS FUNCTION

The most commonly used distribution distance loss functions for knowledge distillation methods in SR are the *Kullback-Leibler* divergence metric (KL) Gou et al. (2021); Fu et al. (2020); Angarano et al. (2022) and the *Mean-squared Error* (MSE). These loss functions are hand-tuned since in the conventional KD methods they are only used once at the final output layers. However, in our progressive KD method we need to calculate the loss at each block level, which makes hand tuning each function impractical. To address this challenge, we use Bayesian Optimization (BO) to automatically optimize the loss functions. Our loss function for block output pairs is:

$$L = \lambda KL(t, s) + (1 - \lambda)MSE(t, s). \tag{6}$$

Whenever we associate original model and output distributions to get the loss values, we run BO to derive the best $\lambda$ value for that anchor point. We use Expected Improvement (EI) Vazquez & Bect (2010) as our acquisition function as it does not require hyperparameter tuning and it is easy for setting intuitive stop conditions. Our $\lambda$ values range between $0.0 - 1.0$ with a $0.01$ granularity. The stopping condition is when the last 20 trials do not improve PSNR value. We run BO for each anchor point until it reaches the stop condition. Then we use the found $\lambda$ value to calculate the loss that we use to train the adaptable model weights.

### 3.3 *AdaSR* TRAINING

To support dynamic runtime environment with constantly changing available resources, it is critical to adapt the model in real time to achieve the most profitable performance while maintaining QoS. One potential approach is to ensemble a set of models and dynamically switch between them. However, this method results in a significantly larger memory footprint, which is infeasible for resource constraint environment such as mobile and IoT devices. To address this challenge, we a training method with shared architecture and weights as outlined in Algorithm 1. For each batch in every epoch, we get the output $t$ of each block of the original model $T$. For each combination of reduced depth and width options $W, D$ of the adapted model $S$, we take the output $s_D^W$ of the same batch. We match the original model's output to the corresponding adapted model's $i^{th}$ block output using Depth Consolidation and Function Matching. We then use Bayesian to set $\lambda$ for Equation 6 to calculate the gradients. Once gradients for all depth and width combinations are derived, we apply them all at once to update the adapted model $S$.

This training method has two advantages: 1) The weights are trained to reduce output distribution discrepancy between the original and adapted models; and 2) applying the gradients together ensures

the final trained weights are shared. Weight sharing allows us to dynamically change the channel and block sizes in real time with no extra memory overhead and less performance impact. Changing the channel and block sizes can be done by bypassing channels and blocks to get the final output on-the-fly to avoid the overhead of reloading models.

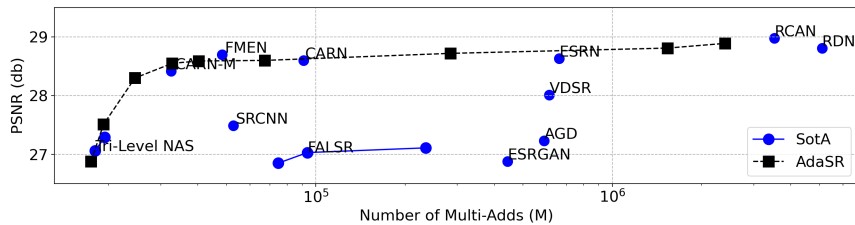

Figure 3: Comparison between our KD-trained models against state of the art against MACs and PSNR. Shown for Set14 at 4x upscaling with patch size 256x256. *AdaSR* is capable of producing models across a range of sizes suitable for different hardware platforms.

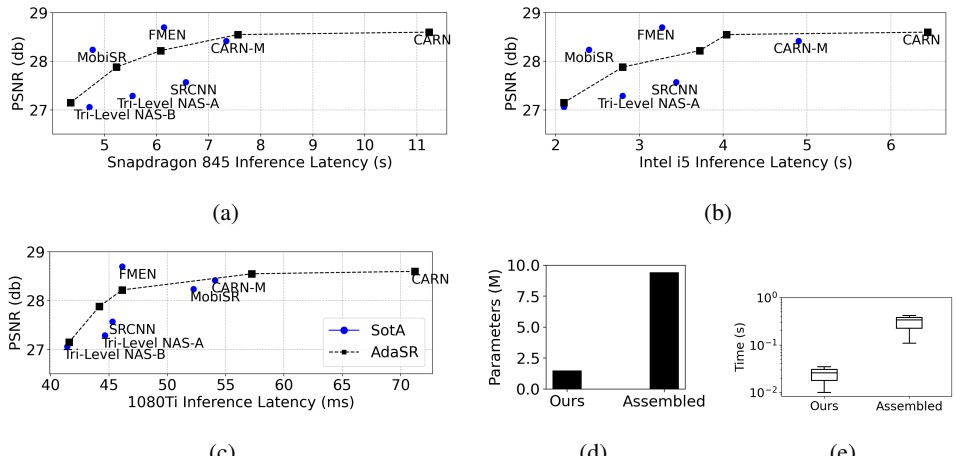

Figure 4: **(a-c)** - Comparison of PSNR vs. inference latency trade-off between *AdaSR* and state-of-the-art models designed for resource constrained deployments. *AdaSR* is developed by changing the depth and width of the original CARN model. Experiment is run on Set14 with 4x upscaling factor for path sizes 256x256. **(d-e)** - Memory cost and interruption time comparison between *AdaSR* and state-of-the-art model assembly.

## 4 EVALUATION

**Experimental Setup.** We use 800 RGB images with patch sizes 64x64 from the standard DIV2K Agustsson & Timofte (2017); Timofte et al. (2018) dataset to train our original models for pre-training. We apply our method on CARN Ahn et al. (2018), ESRGAN Wang et al. (2018b), RCAN Zhang et al. (2018a), and FMEN Du et al. (2022). The pre-training of the original models are done as explained in their corresponding papers. We then freeze our original model and train the adapted models using *AdaSR* with the same dataset. For the adapted models, we use batch size of 16 with standard data augmentations of random rotation, random horizontal flipping, and normalization. We use the ADAM optimizer with $\beta_1 = 0.9, \beta_2 = 0.99$ for all models. We use learning rates for each compact model as described in their corresponding papers, and apply exponential learning rate scheduling. Training till convergence takes between 50 to 200 rounds depending on the model. We train on 4 1080Ti GPUs with batch parallelism. Evaluation is done on Urban100 Huang et al. (2015), Set5 Bevilacqua et al. (2012), Set14 Zeyde et al. (2010), BSD100 Martin et al. (2001), and DIV2k Agustsson & Timofte (2017); Timofte et al. (2018) datasets for both 2x and 4x scaling but present with 4x in the interest of space.

**Model Implementation.** Since we reduce the width and depth of existing a variety of existing models, we denote models as - $[model\ name]\_[depth\ ratio]\_[width\ ratio]$. For example, the original CARN model has 3 block layers and uses 64 cell channel size. So a CARN model with 1 block layer

and 32 channel size is denoted as $CARN\_0.3\_0.5$. We implement our framework in Pytorch with fp16 quantization and deployed with ONNX onn (2022).

Table 1: Comparison between *AdaSR* and state-of-the-art models. Evaluated for 4x SISR and trained on Div2K. Patch size is 256x256.

| Model | Params | FLOPS | Set5 PSNR/SSIM | Set14 PSNR/SSIM | BSD100 PSNR/SSIM | Urban100 PSNR/SSIM |
|---|---|---|---|---|---|---|
| RCAN | 1.6 | 130.2G | 32.63 / 0.900 | 28.87 / 0.788 | 27.77 / 0.743 | **27.01 / 0.814** |
| ESRGAN-Prune | 1.6 | 113.1G | 28.07 / 0.737 | 25.21 / 0.634 | 25.22 / 0.641 | 22.45 / 0.584 |
| AGD-L | 0.90 | 139.4G | **31.86 / 0.892** | **28.40 / 0.801** | **27.47 / 0.724** | 25.55 / 0.695 |
| MDDC-L | **0.79** | **96.6G** | 31.74 / 0.887 | 28.31 / 0.773 | 27.36 / 0.729 | 25.46 / 0.763 |
| **RCAN-1_0.75 (Ours)** | 0.97 | 98.5G | 31.64 / 0.889 | 28.37 / 0.781 | 27.25 / 0.732 | 25.59 / 0.739 |
| CARN-M | 0.4 | 74.32 | **31.92 / 0.890** | 28.42 / 0.776 | 27.44 / 0.730 | 25.63 / 0.768 |
| AGD-M | 0.45 | 110.9G | 30.36 / 0.833 | 27.41 / 0.754 | **27.59 / 0.742** | 24.39 / 0.688 |
| Tri-Level NAS - B | 0.51 | 117.4G | 30.34 / 0.821 | 27.29 / 0.722 | 26.43 / 0.682 | 25.45 / 0.756 |
| MDDC-M | **0.36** | **59.6G** | 31.53 / 0.884 | 28.19 / 0.770 | 27.29 / 0.727 | 25.24 / 0.756 |
| **CARN-0.6_0.5 (Ours)** | 0.39 | 68.5G | 31.60 / 0.878 | **28.46 / 0.759** | 27.34 / 0.717 | **25.68 / 0.783** |
| DRRN | 0.29 | 32.4G | **31.40 / 0.852** | 28.01 / 0.761 | **27.82 / 0.699** | **25.35 / 0.757** |
| Tri-Level NAS - A | 0.24 | 15.4G | 29.80 / 0.753 | 27.60 / 0.748 | 26.22 / 0.679 | 24.77 / 0.666 |
| MDDC-S | 0.25 | 14.8G | 31.31 / 0.879 | 28.04 / 0.767 | 27.19 / 0.723 | 25.03 / 0.747 |
| **FMEN-0.75_0.8 (Ours)** | **0.23** | **14.6G** | 31.38 / 0.874 | **28.09 / 0.773** | 27.16 / 0.731 | 25.11 / 0.714 |

## 4.1 CROSS PLATFORM PARETO OPTIMALITY

First, we evaluate *AdaSR* 's ability to generate models for a wide variety of hardware platforms. We adapt existing models to a range of sizes. We apply our training method to serveral original models, such as CARN Ahn et al. (2018), ESRGAN Wang et al. (2018b), RCAN Zhang et al. (2018a), and FMEN Du et al. (2022). The adapted models are evaluated against state-of-the-art models in Fig. 3. Note that *AdaSR* can generate models for a wide range of MAC values, while most other frameworks are only capable of generating single models. Even though NAS frameworks such as AutoGAN Fu et al. (2020), Tri-Level NAS Wu et al. (2021), and FALSR Chu et al. (2021) can generate multiple models, the range of model sizes are limited by their search spaces. Tri-Level NAS can generate Pareto optimal architectures, but the others have relatively low performance since they are more focused on automating SR model generation than achieving state-of-the-art performance. In comparison, our method can utilize a wide variety of existing architectures and generate a wide range of models with different sizes that are Pareto optimal.

## 4.2 DYNAMIC RUNTIME ENVIRONMENT

Next, we evaluate our framework's ability to create models for dynamic runtime environments. Here we take a single CARN model and use our framework to train it. This generates a single adaptable CARN model that is capable of running inference using various shared operation sizes. We deploy our trained adaptable model along with state-of-the-art models on a desktop GPU (1080Ti), a laptop CPU (i5-5560), and a mobile CPU (Snapdragon 845). We choose these models specifically since they are on the inexpensive end of the SR model space and are designed for deployment on resource constrained hardware. We then compare our adaptable model's performance against the others on inference latency. The results are shown in Fig. 4. It is worth noting that the state-of-the-art models assembled together set the Pareto frontier, while ours is a single model which is capable of adapting to various sizes and can be changed on-the-fly without model reloads.

The results show that our adapted CARN model outperforms hand-designed and NAS-designed models in terms of performance against latency. FMEN Du et al. (2022) is finely tuned for reducing inference latency and is one of the efficient models from the 2022 NTIRE challenge Li et al. (2022), and it outperforms our original model at the beginning. Since FMEN is a block-based GAN, our *AdaSR* can be applied on FMEN as well (results presented later). For MobiSR Lee et al. (2019), it is specifically developed with consideration to mobile CPU compilers and hardware architectures, which is why it performs better on CPU deployments but not on GPUs. Nonetheless, our adapted

CARN model outperforms most efficient models in this size range, and is relatively close to highly efficient hand-designed models. One additional advantage we have over other models is that our model has a single set of weights. Different model sizes simply bypasses channels and blocks during inference and so can be done without any interruptions. In a dynamic runtime environment, available resource for inference fluctuate and so we need to swap among models such that a minimum inference latency is maintained. We can do this by having an assembled set of models and rerouting the execution pipeline to the appropriate models as required. In Fig. 4d, we present the

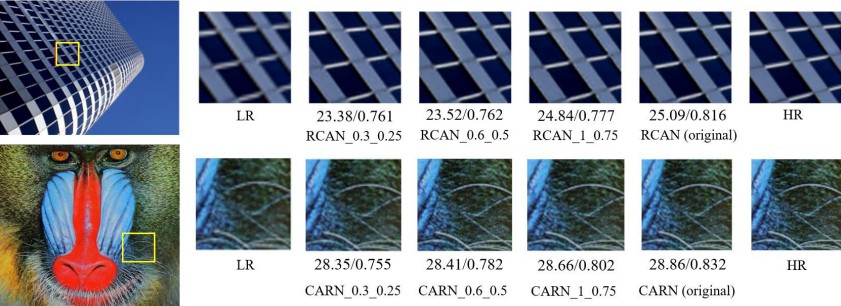

Figure 5: Visual qualitative comparison between *AdaSR* models and the corresponding original models. Images taken from Urban100 (above) and Set14 (below) datasets for 4x resolution with their Low resolution (LR) input and ground-truth High Resolution (HR) samples.

results comparing the total number of parameters of the state-of-the-art models from Fig. 4 assembled together against our single adaptable model. Here, we see clearly that our model is around 20% of the assembled models, so our single model approach is significantly more memory efficient for dynamic runtime environments. Fig. 4e shows the distribution of the amount of time taken to swap between the state-of-the-art models vs. time for our single adaptable model to change between compression levels. The assembled models take magnitudes larger time to load compared to *AdaSR* since they have to swap between models through expensive memory loading/unloading operations from disk, whereas ours is a single model completely loaded into the memory at all times. This causes less interruption of service during runtime. Thus, our approach is much more suitable to dynamic runtime environments than others in current literature.

### 4.3 COMPARISON WITH STATE-OF-THE-ART METHODS

In addition to support adapting models for cross platform deployment and dynamic runtime environment, *AdaSR* achieves state-of-the-art performance (or close to it). In Table 1, we present the comparison between *AdaSR* and state-of-the-art models of similar sizes across a variety of SR benchmark datasets. We group the models into three different size categories based on their number of parameters. Our adaptive training scheme is also capable of generating models without much fine-tuning, unlike works such as AGD which require running search phases for each model size criteria. One interesting observation is that our adaptation of the highly efficient FMEN model also gives highly competitive results, implying that the original model chosen for adaptation is important for the performance of adapted models. This shows the effectiveness of *AdaSR* 's focus on reducing output discrepancy between the original and adapted models. We also present comparison of qualitative results in Fig. 5. Here, we compare the SR outputs of the original models and the outputs generated by *AdaSR* . We observe that as we reduce the sizes of the models progressively, we note a drop in quantitative values as expected, but also that the qualitative difference is not significant and even our smallest models provide large improvements over the LR images.

### 5 CONCLUSION

In this paper, we propose *AdaSR* , which can be used to adapt existing SR models for different hardware and adaptively change the compute graph in dynamic runtime environment, achieved by changing the depth and the channel size in real time on a single architecture with no extra cost on memory and/or storage. We perform shared weight training using a progressive training approach to reduce output discrepancies between the original and adapted model with a combination of function matching, max-norm regularization, Bayesian-tuned loss functions, and gradient aggregated training to improve training performance. Extensive tests on a variety of hardware and datasets show that *AdaSR* has Pareto optimal performance, reduced memory footprint, and supports real-time adaption in dynamic runtime environments.

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
