# OpenReview forum: "AdaSR: Adaptive Super Resolution for Cross Platform and Dynamic Runtime Environments"
_ICLR.cc/2024/Conference — Submitted to ICLR 2024_

### Official Review · Reviewer_vLeU · 2023-10-29

**Soundness:** 3 good
**Presentation:** 2 fair
**Contribution:** 3 good
**Rating:** 5
**Confidence:** 3

**Summary:**

In this paper, the authors present a framework for training an SR model that delivers consistent performance across different platforms. The techniques such as Block-level Progressive Knowledge Distillation, Function Matching, Depth Consolidation, and Bayesian-tuned Loss Function were utilized. This approach achieved the good trade-off curve in PSNR relative to computational cost, and the performance was validated across a range of devices.

(+) Typo or Layout Error
- Section 3.2.2 : "have the same dimensions/"
- Caption of Figure 2 : overlapped with the main paragraph

**Strengths:**

- The problem setup is commendable, addressing the important issue of super-resolution across different platforms and dynamic runtime environments.
- A variety of experiments support the claims made in the paper.
  - Figure 3 shows the superiority of the trade-off curve compared to conventional knowledge distillation models.
  - Figure 4 demonstrates operation on various architectures such as Snapdragon 845, Intel i5, and RTX 1080 Ti.
  - Table 1 compares the results on different datasets.

**Weaknesses:**

- In Figure 4 and Table 1, the performance of AdaSR is still inferior to that of MobiSR and the large SR model.
- The training process of AdaSR appears to be complex, making it more challenging to extend to other platforms compared to designing a bespoke SR model for the target platform.

**Questions:**

- Why regularization for mapping layer is enough to keep the inference performance when removing the mapping layer? In my experience SR model is very sensitive to that kind of operation. Is there a related analysis or experiment?

---

> ### Author Response · Authors · 2023-11-23
>
> **1.** The purpose of our paper is not to beat state-of-the-art, but to achieve comparative performance while being able to dynamically compress a single model according to a device’s need. We will clarify this in the paper.
>
> **2.** The main advantage of this training method is that it allows us to train a single model to be capable of performing inference at various latencies without significant drop in performance. To our knowledge, no other work attempts to build such a model. Our motivation is slightly different in that instead of targeting a specific device with a model, we train a single model that can be used on any targeted devices within a certain hardware scope. We will clarify this in the paper.
>
> **3.** Thank you for pointing this out. Correct, we also found the SR model is sensitive to changes in removal of this layer. We found that using regularization to make the mapping layer to have very small weights somewhat mitigates this issue. We still observe a drop in performance compared to having the mapping layer. However, having no regularization or too large regularization thresholds can result in a much more significant drop in performance. The threshold we used has been tuned to make the performance drop to what we feel are within acceptable limits. We will add these details in the paper.

---

### Official Review · Reviewer_w1MH · 2023-10-31

**Soundness:** 2 fair
**Presentation:** 3 good
**Contribution:** 2 fair
**Rating:** 5
**Confidence:** 3

**Summary:**

Super-resolution models significantly enhance the visual quality of low-resolution images. However, a notable limitation is their challenging adaptability to different hardware platforms, given the platform diversity. Moreover, these models often lack consideration for the runtime environment in which they operate. This environment can substantially impact their performance, influenced by both the hardware characteristics and available runtime resources. In response to these limitations, this paper introduces AdaSR, a solution designed to address these challenges. AdaSR employs a progressive knowledge distillation model training approach, which optimizes memory usage by dynamically adjusting depth and channel sizes based on the specific hardware during runtime while maintaining accuracy as much as possible. The experimental results demonstrate the effectiveness of AdaSR to a certain extent.

**Strengths:**

+ The motivation of this paper is insightful.
+ The design of the approach is clear and straightforward.
+ The paper is well organized.

**Weaknesses:**

- Some key procedures of this approach need further clarification.
- Additional experiments are necessary to assess the model's performance in dynamic runtime environments.

**Questions:**

1. This approach requires retraining of existing models. What are the associated training costs in terms of time and hardware expenses?
2. In Chapter 3.2.1, it is mentioned, “... increase the size of the adaptable model, ... and repeat the process ...”. The question is: When increasing the size each time, are all blocks in the adaptable model synchronized? For instance, do block 1 through block M increase to the same block size and channel size each time? If this is the case, does this approach iterate through all possible solutions within the search space?
3. Is it feasible to apply or extend this approach to non-block-based models?
4. In the experimental setup, the parameters beta_1 and beta_2 for the ADAM optimizer are set to the same values as those in the FAKD approach. Are there specific reasons for maintaining these parameters consistent with FAKD?
5. It would be beneficial to include information about the available memory spaces in the experiments, as it is a crucial factor in characterizing the dynamic runtime environment.
6. In Table 1, the significance of the bold values is ambiguous. They do not represent the best performance in the comparisons. More detailed clarifications are required.
7. Chapter 4.2 does not sufficiently demonstrate AdaSR's robust adaptability in a dynamic runtime environment. An ablation study may be necessary, as it is an important statement declared in the Introduction. This is particularly relevant for showcasing AdaSR's adaptability in a dynamic runtime when “other running applications are present.”
8. A minor writing issue: In Chapter 3.3, there is a phrase, “... we a training method with ...”

---

> ### Author Response · Authors · 2023-11-23
>
> We thank the reviewer for the valuable comments and feedback. We address the concerns as follows -
>
> **1.** Since we perform layer-wise training, we undergo more back-propagation iterations than if we were to train the full model at once. The total number of iterations would be *N* x *C* x *e* where *N* is the number of blocks, *C* is the number of channel sizes. and *e* the number of epochs till convergence. However, the relative cost of each backpropagation is cheaper since we freeze all previous layers, and only train one layer at a time. To give an estimate, our base CARN model requires 8 GPU days to fully train to convergence on a 1080Ti GPU. We will add these details to our paper.
>
> **2.** Yes, we iterate through each possible solution. However, we can reduce this exploration space depending on what devices we may possibly target. We will add this detail to the paper.
>
> **3.** At this stage we have only worked on block based models. Applying this method will be challenging to non-block based models since the granularity at which layers can be progressively distilled across needs to be determined first, which is an extra research question to address. However, if that can be done, there is a possibility of applying this method to non-block based architectures as well.
>
> **4.** We tried several other parameters, and this gave the overall best results. That said, this had little performance impact compared to others.
>
> **5.** This is a good point, we will add this in our results.
>
> **6.** Thank you for pointing this out. We will remove the bold font from our model names.
>
> **7.** Thank you for pointing this out. We will add more ablation studies on this.
>
> **8.** Thank you, we will fix this.

---

### Official Review · Reviewer_DMaG · 2023-10-31

**Soundness:** 2 fair
**Presentation:** 2 fair
**Contribution:** 1 poor
**Rating:** 3
**Confidence:** 4

**Summary:**

The authors proposed AdaSR, a framework that can train multiple image super resolution models via shared architecture and weights for cross platform deployment and dynamic runtime environment.

**Strengths:**

1. The technical part of this paper is written clearly and easy to follow.

2. The AdaSR achieves fairly good results, but I'm not sure if they are SOTA since the comparisons involve parameters, FLOPS and multiple SR datasets that are hard to align. Therefore, I suggest the author provide a more comprehensive experiment report.

**Weaknesses:**

1. The novelty of this paper is quite limited. It seems that AdaSR adapts the width and depth. It's pretty similar to many previous works such as [*]. The only novelty I could find in AdaSR is to combine progressive distillation and NAS together.

2. The performance improvement is marginal compared with MDDC. It's hard to tell which method is better based on Table 1. I suggest the author provide a more comprehensive experiment report. Also, the authors should compare with more recent baselines. By the way, I couldn't find any citations in your paper about MDDC in Table 1.

3. Lack of experiment results on perceptual metrics. I suggest the authors report LPIPS results.

4. Lack of visual qualitative comparison. I can only find two visual qualitative comparisons in figure 5.

5. The authors use tons of indentation between figures, paragraphs and headings. I'm not sure if this kind of behavior violate ICLR instructions.

[*] Compiler-aware neural architecture search for on-mobile real-time super-resolution, ECCV 2022.

**Questions:**

I couldn't find any citations in your paper about MDDC in Table 1.

---

> ### Author Response · Authors · 2023-11-23
>
> We thank the reviewer for the valuable comments and feedback. We address the concerns as follows -
>
> **1.** Our main novelty is that we can do this in a single model, without having to create and train a new model for every different platform. We will clarify this in the paper.
>
> **2.** The purpose of our paper is not to beat state-of-the-art, but to achieve comparative performance while being able to dynamically compress a single model according to a device’s need. We will clarify this in the paper.
>
> **3.** Thank you for pointing this out. We will add them to the Appendix
>
> **4.** We have more such results, but removed them due to space limitations. We will add them in the Appendix.
>
> **5.** Thank you for clarifying this, we will re-format where required.

---

### Official Review · Reviewer_eni7 · 2023-11-03

**Soundness:** 2 fair
**Presentation:** 3 good
**Contribution:** 3 good
**Rating:** 5
**Confidence:** 4

**Summary:**

The paper proposes an adaptive SR framework, AdaSR, for cross-platform deployment and dynamic runtime environment. AdaSR can be implemented in existing SR models and achieves a promising tradeoff between latency and performance.

**Strengths:**

+ This is well-motivated for practical applications of SR models. The paper is easy to follow.
+ The proposed method can achieve a good balance between latency and performance.

**Weaknesses:**

Although I greatly appreciate the motivation for practical applications of SR methods in diverse platforms, my main concern is about the empirical evaluation of the effectiveness.

- As claimed in the paper, "none ... address the challenges in dynamic runtime environment ..." and the proposed AdaSR aims to address this issue. It is confusing and ambiguous for the experiment settings for the cross-platform and dynamic runtime environment.

- The evaluations for "cross-platform Pareto optimality" (Sec. 4.1) and "dynamic runtime environment" (Sec. 4.2) are conducted on a very small test set, i.e., set14, Fig.3 and Fig.4. I do not think those results are convincing.

- In Sec. 4.3, "to support adapting models for cross-platform deployment and dynamic runtime environment, AdaSR achieves state-of-the-art performance, AdaSR achieves state-of-the-art performance". However, Tab.1 just shows the results on existing SR datasets. How can we learn from those evaluation results and how to demonstrate the effectiveness under cross-platform deployment and dynamic runtime environment?

**Questions:**

Please refer to the issues in "Weakness" section.

---

> ### Author Response · Authors · 2023-11-23
>
> We thank the reviewer for the feedback. Our responses are as follows.
>
> **1.** Thank you for this feedback. Our claim of being suitable for dynamic runtime environments hinges on the fact that our model can change its inference latency on the fly without requiring the reloading of models from disk to memory. We will clarify this in the paper. For our experiments for cross-platform compatibility, we demonstrate that a single model can perform at various latencies across various platforms without the need to reload new models (see Figure 4). This is not achieved by the state-of-the-art we compare against. For demonstration of dynamic runtime effectiveness, in Figure 4 that our model can achieve a variety of latencies in the same session dynamically. We will work on making these points clearer in the paper.
>
> **2.** We perform our experiments on a variety of test sets, as shown in Table 1. We limit to plotting only the graphs in Figure 3 due to space. We will add the rest of the experimental results in the Appendix.
>
> **3.** This section is for directly comparing model performance to similarly sized models. We show here that our method can effectively reduce the model sizes with minimal loss of performance since it is competitive with other state-of-the-art.

---

### Meta-Review · Area_Chair_NLky · 2023-12-10

**Metareview:**

The authors propose an Adaptive SR framework (AdaSR) via shared architecture and weights for cross platform deployment and dynamic runtime environment. With AdaSR promising tradeoffs are achievable between latency and performance while using prior SR models.

The reviewers point out multiple weaknesses and issues:

- Reviewer eni7 questions the empirical evaluation of the effectiveness
- Reviewer DMaG finds the novelty "quite limited", marginal performance improvements, missing comparisons with more recent baselines, a lack of experimental results
- Reviewer w1MH requires further clarification for some key procedures and additional experiments in dynamic runtime environments.
- Reviewer vLeU considers that AdaSR leads to inferior performance wrt MobiSR and large SR model and has a complex training process

The authors provide responses but they are not convincing enough for the reviewers to recommend acceptance.

We invite the authors to benefit from the received feedback and further improve their work.

**Justification For Why Not Higher Score:**

The paper does not meet the acceptance bar due to the multiple weaknesses and issues pointed out by the reviewers.

**Justification For Why Not Lower Score:**

N/A

---

### Decision · Program_Chairs · 2024-01-16

Reject